# A Joint Model Based on Post-Treatment Longitudinal Prognostic Nutritional Index to Predict Survival in Nasopharyngeal Carcinoma

**DOI:** 10.3390/cancers16051037

**Published:** 2024-03-03

**Authors:** Po-Wen Hsiao, Yu-Ming Wang, Shao-Chun Wu, Wei-Chih Chen, Ching-Nung Wu, Tai-Jan Chiu, Yao-Hsu Yang, Sheng-Dean Luo

**Affiliations:** 1Department of Otolaryngology, Kaohsiung Chang Gung Memorial Hospita, Chang Gung University College of Medicine, Kaohsiung 833, Taiwan; a8011@cgmh.org.tw (P.-W.H.); jarva@cgmh.org.tw (W.-C.C.); taytay@cgmh.org.tw (C.-N.W.); 2Department of Radiation Oncology & Proton and Radiation Therapy Center, Kaohsiung Chang Gung Memorial Hospital, Chang Gung University College of Medicine, Kaohsiung 833, Taiwan; scorpion@cgmh.org.tw; 3School of Traditional Chinese Medicine, College of Medicine, Chang Gung University, Taoyuan 333, Taiwan; r95841012@cgmh.org.tw; 4Graduate Institute of Clinical Medical Sciences, College of Medicine, Chang Gung University, Taoyuan 333, Taiwan; sean1207@cgmh.org.tw (S.-C.W.); kuerten@cgmh.org.tw (T.-J.C.); 5Department of Anesthesiology, Kaohsiung Chang Gung Memorial Hospital, Chang Gung University College of Medicine, Kaohsiung 833, Taiwan; 6Division of Hematology-Oncology, Department of Internal Medicine, Kaohsiung Chang Gung Memorial Hospital, Chang Gung University College of Medicine, Kaohsiung 833, Taiwan; 7Department of Traditional Chinese Medicine, Chang Gung Memorial Hospital, Chiayi 613, Taiwan; 8Health Information and Epidemiology Laboratory of Chang Gung Memorial Hospital, Chiayi 613, Taiwan

**Keywords:** nasopharyngeal carcinoma, prognostic nutritional index, joint model, post-treatment, follow-up, overall survival

## Abstract

**Simple Summary:**

A low prognostic nutritional index (PNI) is linked to poor survival in patients with nasopharyngeal carcinoma (NPC), but existing research primarily examines the pre- or post-treatment PNI at single timepoints. Our study employs joint modeling to investigate the relationship between longitudinal PNI data from routine visits and overall survival. This approach addresses biases inherent in traditional time-varying covariate Cox models. Our findings indicate that decreased PNI levels during follow-up correlate with a reduced overall survival. Specifically, a post-treatment PNI below 38.1 markedly increases the risk of 90-day mortality. This emphasizes the importance of routine longitudinal PNI data in predicting survival outcomes for patients with NPC, offering a comprehensive perspective compared to isolated timepoint measurements.

**Abstract:**

Background: a low PNI in patients with NPC is linked to poor survival, but prior studies have focused on single-timepoint measurements. Our study aims to employ joint modeling to analyze longitudinal PNI data from each routine visit, exploring its relationship with overall survival. Methods: In this retrospective study using data from the Chang Gung Research Database (2007–2019), we enrolled patients with NPC undergoing curative treatment. We analyzed the correlation between patient characteristics, including the PNI, and overall survival. A joint model combining a longitudinal sub-model with a time-to-event sub-model was used to further evaluate the prognostic value of longitudinal PNI. Results: A total of 2332 patient were enrolled for the analysis. Separate survival analyses showed that longitudinal PNI was an independent indicator of a reduced mortality risk (adjusted HR 0.813; 95% CI, 0.805 to 0.821). Joint modeling confirmed longitudinal PNI as a consistent predictor of survival (HR 0.864; 95% CI, 0.850 to 0.879). An ROC analysis revealed that a PNI below 38.1 significantly increased the risk of 90-day mortality, with 90.0% sensitivity and 89.6% specificity. Conclusions: Longitudinal PNI data independently predicted the overall survival in patients with NPC, significantly forecasting 90-day survival outcomes. We recommend routine PNI assessments during each clinic visit for these patients.

## 1. Introduction

Nasopharyngeal carcinoma (NPC) is an epithelial malignancy that originates from the nasopharyngeal mucosa, exhibiting an unequal geographic distribution [1]. In 2020, there were about 133,000 new cases of NPC and 80,000 deaths globally, with >70% of cases in East and Southeast Asia [2]. The pathogenesis of nasopharyngeal carcinoma (NPC) has been traditionally linked to Epstein–Barr virus (EBV) infection, genetic predispositions, and environmental factors [1]. Building on this, recent theories suggest framing NPC as a pathological ecosystem, where disease progression is influenced by a complex network of ecological and evolutionary interactions, involving the dynamic nature of the cancer and its capacity to adapt within the biological landscape it inhabits [3]. Radiotherapy is the primary treatment for NPC, with concurrent chemoradiotherapy (CCRT) offering survival benefits for the advanced stage of this disease [4]. Despite advances in treatments, many patients with NPC continue to experience unfavorable survival outcomes, particularly those with advanced disease, with 5-year overall survival rates of approximately 70% [5,6]. However, traditional prognostic parameters have a limited capacity to predict treatment outcomes and survival in patients with varying baseline health conditions, which can change over the course of treatment. Hence, there is a need to develop a novel predictor to aid clinicians in identifying patients at risk and optimizing individualized care.

Many studies have shown that systemic inflammation and nutritional status significantly impact survival in various cancer types [7,8,9,10]. In 1980, Buzby and colleagues were the first to introduce the prognostic nutritional index (PNI) as a tool to evaluate the nutritional status and predict the risk of surgical complications [11]. The PNI is determined by assessing the serum albumin levels and the total lymphocyte count in the peripheral blood and has demonstrated prognostic value in various types of malignancies, including lung cancer, esophageal cancer, gastric cancer, colorectal cancer, pancreatic cancer, liver cancer, as well as head and neck cancer [12,13,14,15,16,17,18]. For patients with NPC, increasing evidence has demonstrated that a low PNI is associated with poorer survival outcomes in both the short and long term [19,20,21,22,23,24,25]. Previous research has predominantly focused on the pre-treatment PNI, although some studies have also examined the post-treatment levels [26,27]. Whether evaluating pre- or post-treatment PNIs, these studies have relied on isolated measurements taken at specific timepoints, neglecting the extensive longitudinal PNI data gathered during routine post-treatment surveillance. This represents a missed opportunity to utilize valuable data for predicting clinical outcomes during regular follow-up appointments.

Joint modeling was adopted in this study to utilize PNI data gathered from routine patient surveillance, aiming to enhance survival outcome analysis. This model combines longitudinal and time-to-event data into a unified framework, allowing for simultaneous analyses. The early development of joint modeling was primarily motivated by the challenges encountered in AIDS research [28,29,30,31,32]. In recent decades, it has found widespread application in diverse areas, especially in cancer studies which explore the relationship between longitudinal prognostic parameters and time-to-event outcomes [33,34,35,36]. The key advantage of this method lies in its ability to handle covariates that are measured irregularly or are subject to measurement errors [37,38].

Thus, the aim of this study was to use joint modeling to investigate the relationship between the longitudinal PNI trajectory and the overall survival in patients with NPC. This marks the first study to explore this association using this particular approach. Additionally, we sought to identify an optimal PNI cut-off value, assisting clinicians in dynamically predicting the individualized survival of patients.

## 2. Materials and Methods

Patient Selection and Data Extraction: This study was approved by the institutional review board of the Chang Gung Medical Foundation (reference number: 202001136B0C601). Given its retrospective and observational nature, the need for informed consent was waived. A cohort study was carried out through a retrospective review of medical records from the Chang Gung Research Database (CGRD), a multi-institutional medical database in Taiwan. The CGRD sources data from the Chang Gung Memorial Hospital (CGMH), encompassing seven medical institutes across the country, which collectively admit over 280,000 patients annually. All patients diagnosed with nasopharyngeal carcinoma (NPC) between January 2007 and December 2019 were identified based on their registration in the Taiwan Cancer Registry.

The exclusion criteria were as follows: patients presenting with non-typical nasopharyngeal carcinoma pathologies; those with distant metastasis at diagnosis; and those with a prior registration of nasopharyngeal carcinoma. In addition, patients who did not receive therapy for nasopharyngeal carcinoma with curative intent, those with another malignancy, or those with incomplete body mass index (BMI) and protein-nutrition index (PNI) data were also excluded from this study.

Demographic data including age, sex, diabetes mellitus (DM), hypertension (HTN), BMI, and overall survival (OS) were collected and analyzed. The OS was defined as the time from the diagnosis to death from any cause or censoring at the last follow-up. The treatment protocol included intensity-modulated radiotherapy (IMRT) and concurrent chemotherapy with or without induction chemotherapy. The American Joint Committee on Cancer TNM staging system (7th and 8th edition) was employed for the classification of NPC stages.

Determination of the PNI Cut-Off Value: The PNI was defined as 10 × serum albumin (g/dL) + 0.005 × total lymphocyte count (count/μL), with the formula originating from the work of Onodera et al. [39]. Each set of PNI data was collected and calculated using individual peripheral blood routine tests throughout the entire course of the disease, starting from the point at which the NPC diagnosis had been confirmed. The optimal cut-off value was further determined by receiver operating characteristic (ROC) curves, which utilized the PNI data which had been collected for each patient 90 days prior to either the patient’s time of death or their last documented medical visit. The OS served as the outcome measure for the ROC analysis.

Development of the Joint Model: The joint modeling approach was employed to estimate the relationship between the overall survival and the longitudinal change in the PNI. Özgür Asar et al. have provided an intuitive and comprehensive tutorial framework demonstrating how to combine a linear mixed-effect sub-model with a survival sub-model to form a joint model [38]. The first step was to build a linear mixed-effect model, using repeated measurements of PNI data as the response variables. This phase ignored the potentially informative nature of the censoring of each PNI sequence due to the occurrence of a survival event. In the second step, a separate survival analysis was conducted, fitting a multivariate Cox proportional hazards model with covariates including age, sex, DM, HTN, BMI, cancer staging, and treatment protocol, with OS as the event. Here, longitudinal PNI observations were treated as time-varying covariates, with each observed PNI value being carried forward at a constant level until the following measurement. For the final stage of this analysis, we engaged in a joint analysis of longitudinal PNI and survival outcomes. During this stage, the current (unobserved) PNI values were incorporated into the survival sub-model.

Statistical Analysis: All statistical tests were two-tailed, with a *p*-value < 0.05 being considered statistically significant. Patient baseline demographics and clinical features were categorized based on outcome status and subsequently subjected to t-tests for continuous variables, while the categorical variables were examined via Chi-square tests /Fisher’s exact tests to elucidate differences between the two outcome groups (alive or deceased). Joint modeling was conducted using the JM package from R version 3.6 (R Center for Statistical Computing, Vienna, Austria). The receiver operating characteristic (ROC) curve analysis employed the pROC package and the rio package, with Youden’s index being used to establish the optimal cut-off for the PNI values. All other statistical computations were conducted using SAS, version 9.4 (SAS Institute, Cary, NC, USA).

## 3. Results

### 3.1. Patient Enrollment and Demographics

This cohort initially included 3269 patients diagnosed with NPC. Exclusions were made for 81 patients with non-typical NPC pathologies, 267 patients with distal metastasis, 9 with a previous history of NPC, and 89 who did not undergo treatment with curative intent, leaving 2823 patients. Further exclusions were applied to 220 patients lacking PNI data, 48 without BMI data, and 223 presenting with malignancies other than NPC. Subsequently, the remaining 2332 patients were selected for further analysis (Figure 1).

The median follow-up time was 5.51 years for patients who had been alive during the observed period and 2.87 years for patients who were dead. During the follow-up period, 638 patients (27.4%) died. Compared to the patients who remained alive (1694 patients [72.6%]), those in the mortality group were characterized by an older mean age (54.4 ± 13.2 vs. 48.1 ± 11.3 years), a greater predominance of male patients (79.9% vs. 73.5%), a more advanced disease status (87.3% vs. 66.1% with *AJCC* stage III and IV), and a lower mean BMI (24.5 ± 4.2 vs. 25.1 ± 4.0 kg/m^2). Additionally, the mortality group had a higher prevalence of both DM and hypertension (DM: 6.9% vs. 3.8%; HTN: 11.8% vs. 7.2%). Nearly 90% of patients received concurrent chemoradiotherapy with or without induction chemotherapy (2090 patients [89.6%]), and intensity-modulated radiotherapy (IMRT) alone was administered to 242 patients (10.4%). No statistically significant difference in terms of treatment protocol was observed between the group of patients who survived and the group who did not (Table 1).

### 3.2. Joint Modeling

In the first step, a random-intercept-and-random-slope linear mixed-effects model was employed for the analysis of longitudinal data. This revealed that the PNI decreased with the increasing time of follow-up (estimate = −1.568; 95% confidence interval [CI], −1.784 to −1.352) and with the age of the patient at enrollment (estimate = −0.117; 95% CI, −0.132 to −0.101). Male patients tended to have a higher PNI compared to the female patients (estimate = 0.734; 95% CI, 0.311 to 1.158), and the patients with DM were associated with a lower PNI (estimate = −1.818; 95% CI, −2.752 to −0.885). Moreover, the patients with a higher BMI showed increased PNI levels (estimate = 0.086; 95% CI, 0.040 to 0.131). There was a notable trend of decreasing PNI levels as the stage of NPC advanced. In terms of treatment, the patients who received CCRT (estimate = 1.606; 95% CI, 0.843 to 2.369) and those undergoing induction chemotherapy followed by CCRT (estimate = 1.492; 95% CI, 0.597 to 2.387) exhibited a higher PNI compared to those treated with IMRT alone (Table 2).

Secondly, a time-varying covariate Cox proportional hazards model was applied to the survival data, showing that time-varying PNI was independently associated with a reduced risk of mortality (adjusted hazard ratio [HR], 0.813; 95% CI, 0.805 to 0.821). Age and male gender were linked to worse survival outcomes, with adjusted HRs of 1.011 (95% CI, 1.005 to 1.018) and 1.445 (95% CI, 1.188 to 1.759), respectively (Table 3).

In the final step, merging these two sub-models led to the formation of the joint model. Within this, the survival sub-model indicated that longitudinal PNI remained an independent predictor of survival outcomes, accounting for potential bias due to measurement errors. It was found that a one-unit rise in PNI corresponded to a 0.864-fold reduction (95% CI, 0.850 to 0.879) in the risk of all-cause mortality among patients with NPC (Table 4).

### 3.3. Cut-Off Value Determination

To assess the prognostic utility of PNI in patients with NPC, PNI data were collected 90 days before either the recorded date of death or the last available clinical visit for each patient. The ROC analysis identified an optimal PNI cut-off of 38.1 (sensitivity: 90.0%; specificity: 89.6%), with an area under the ROC curve (AUROC) of 0.949 (95% CI, 0.937 to 0.961) (Figure 2) The PNI distribution of the two groups (survival lasting more than 90 days vs. under 90 days post final PNI evaluation) is depicted as a box-plot in Figure 3. A significant difference was observed between the groups (Mann–Whitney U test, *p* < 0.001).

## 4. Discussion

Given the limitations of the traditional TNM staging system for NPC, there is a pressing need for alternative prognostic markers that can offer a more precise and individualized prediction of patient outcomes. The prognostic nutritional index, which assesses both nutritional and immunological status, has emerged as a notable candidate. The prognostic significance of the PNI has been demonstrated by a series of studies, but the vast majority of these have relied solely on single-timepoint pre-treatment PNI measurements [19,20,21,22,23,24,25]. To the best of our knowledge, this is the first study to utilize longitudinal PNI data for outcome prediction in patients with nasopharyngeal carcinoma.

Our study utilized the Chang Gung Research Database (CGRD), a comprehensive multi-medical institutional database with extensive coverage of overall and disease-specific data in Taiwan. Research using the CGRD is recognized for its high quality and has contributed to healthcare advancements in Taiwan [40,41]. In our analysis, reduced PNI levels during follow-up were associated with a decrease in the overall survival. Specifically, each one-unit increase in PNI resulted in a 0.864-fold decrease (95% CI, 0.850 to 0.879) in the risk of all-cause mortality. Furthermore, a PNI below 38.1 significantly raised the risk of mortality within the following 90 days, with 90.0% sensitivity and 89.6% specificity. Considering the PNI’s accessibility, cost-effectiveness, and its notable role in forecasting patient outcomes, we recommend adopting the PNI in regular follow-up appointments to enhance the prognostic risk assessment of individual patients. For patients exhibiting low PNI levels, prompt nutritional interventions and tailored treatment strategies may be crucial.

Additionally, our longitudinal analysis also revealed that male patients with NPC had higher PNI levels, consistent with findings from two large meta-analyses [20,21]. However, despite a higher PNI, the male patients in our study showed poorer outcomes in the survival model, suggesting that factors other than the PNI affect gender-based survival differences in NPC. OuYang et al. reported that female patients with NPC generally had better survival rates, even after adjustments for factors including BMI, smoking, drinking, and disease severity. This survival advantage is thought to be linked to hormonal differences, especially considering the diminished benefit among postmenopausal women [42].

The PNI, derived from serum albumin and lymphocyte counts, can decrease due to hypoalbuminemia or lymphocytopenia. These components are key to exploring the connection between a low PNI and reduced survival in patients with NPC. Serum albumin is recognized as a biomarker for evaluating nutritional and inflammatory statuses in patients with cancer [43,44]. In NPC, which is predominantly managed with chemoradiotherapy (CRT), malnutrition is further aggravated by complications such as severe mucositis and gastrointestinal reactions [45]. In addition, hypoalbuminemia can impair the human immune system, compromising both cellular and humoral immunity. This compromise elevates the risk of infections and can trigger further systemic inflammatory responses, thereby promoting cancer progression [46,47,48]. Furthermore, systemic inflammatory markers like C-reactive protein, interleukin-1, and the tumor necrosis factor are intricately linked with hypoalbuminemia due to their role in inhibiting albumin synthesis [49]. Lymphocytes, integral to immunosurveillance, including tumor detection and destruction, also contribute to tumorigenesis prevention through cytokine production [50,51]. Additionally, lymphocytopenia has been linked to reduced chemotherapy efficacy in patients with cancer [52]. Thus, it is unsurprising that lymphocytopenia has been shown to be indicative of unfavorable outcomes [53,54]. Considering these aspects, the PNI serves as a comprehensive indicator for predicting the outcomes of patients with NPC by assessing their nutritional state, systemic inflammatory response, and overall immune health.

Currently, only four studies have explored the prognostic value of post-treatment PNI values. The first study, involving 23 patients with head and neck cancer undergoing concurrent chemoradiotherapy (CCRT), observed a correlation between the PNI levels during treatment and the severity of mucositis. It further proposed utilizing the post-treatment PNI as a criterion to guide discharge timing after completing CCRT [55]. The second study, which included 124 patients with head and neck cancer—20 of whom had NPC—found that low post-treatment PNI levels significantly indicated poor prognoses [56]. The third study, specifically targeting patients with NPC, linked lower PNI levels immediately post treatment with poorer survival outcomes [26]. Similarly, the fourth study, assessing the PNI one month post treatment in patients with NPC, confirmed this association. Importantly, it highlighted that dynamic PNI changes over time are more predictive of prognosis, emphasizing the importance of tracking PNI changes over time rather than relying solely on the pre-treatment levels [27]. Our study enhances these findings by analyzing a comprehensive dataset of longitudinal PNI from routine post-treatment surveillance, instead of a single measurement. With over two thousand patients enrolled, it represents the largest cohort and may, therefore, provide a more robust assessment of PNI’s prognostic value in NPC survival.

In our study, we employed joint modeling to explore the relationship between post-treatment longitudinal PNI and survival outcomes. Traditionally, such analyses utilized the time-varying covariate Cox model (TVCM), incorporating repeated measurements of the PNI as time-varying covariates into a Cox proportional hazards model. However, this requires that the covariates be external factors, suggesting that their future values are predetermined and remain unaffected by the occurrence or non-occurrence of the event. Also, in this approach, the last-observation-carried-forward (LOCF) method is typically used, as marker observations are available only at discrete intervals. This can be a significant drawback, potentially introducing bias due to the continuous nature of the biomarker. Moreover, this method assumes that these covariates are free from measurement errors [57,58,59]. Given the characteristics of longitudinal PNI data, we found joint modeling to be more appropriate for our analysis. Our results indicated that the longitudinal PNI remained a significant predictor of the overall survival, even when considering potential biases from LOCF and measurement errors, which were addressed by the joint modeling methodology.

This study has several limitations. Firstly, with it being a retrospective study, our findings are susceptible to bias. Additionally, all the participants were from Taiwan, raising questions about the applicability of our results to other populations. Secondly, while markers of EBV infection are crucial covariates in NPC research, they were excluded from our study as they were not routinely measured in earlier cases. Thirdly, there is a potential bias due to the use of different staging systems over time; earlier patients were classified using the *AJCC 7th edition*, whereas later patients were assessed with the *8th edition*. Finally, our study may not have comprehensively considered all the factors impacting the survival of patients with NPC. Given the substantial prognostic significance of 90-day survival revealed in our analysis, further research is needed to investigate these underlying factors.

## 5. Conclusions

The longitudinal PNI data served as an independent marker for predicting the overall survival in patients with NPC. The post-treatment PNI levels could significantly predict survival in the following 90 days. Given its cost-effectiveness and convenience, we advise regular PNI measurements during each clinic visit for patients with NPC. For those at a high risk, nutritional interventions or adjustments in therapy strategies may be necessary.

## Figures and Tables

**Figure 1 cancers-16-01037-f001:**
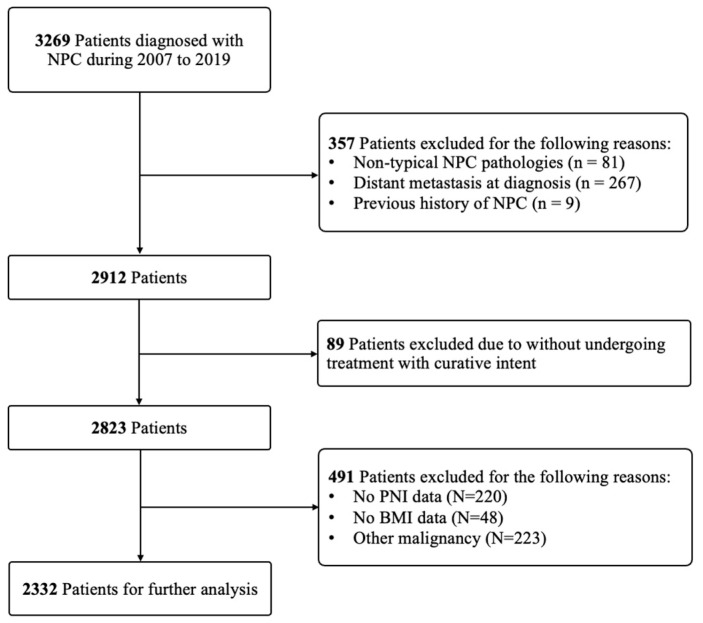
Flowchart of the patients included in this study and analysis.

**Figure 2 cancers-16-01037-f002:**
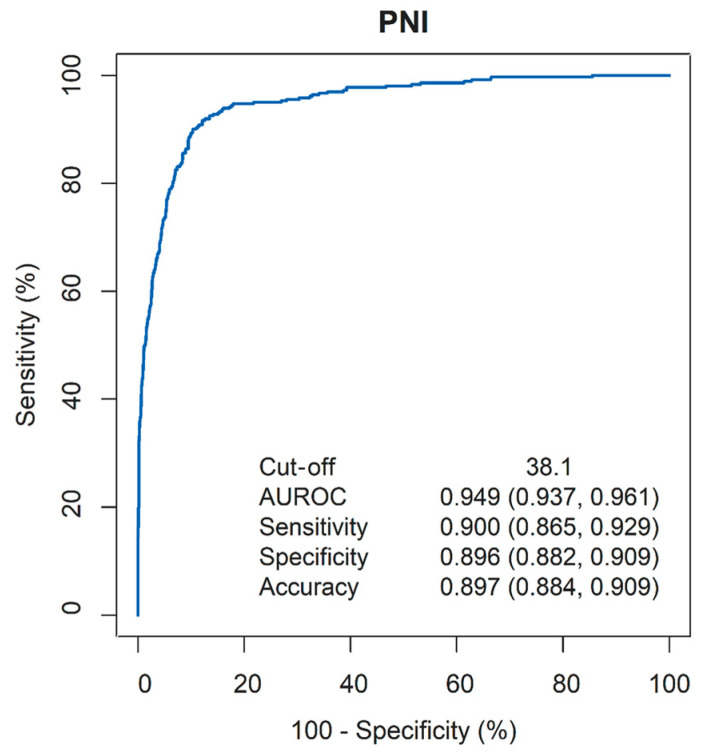
Receiver operating characteristic (ROC) curves for overall survival (OS) according to the PNI 90 days before death or last visit.

**Figure 3 cancers-16-01037-f003:**
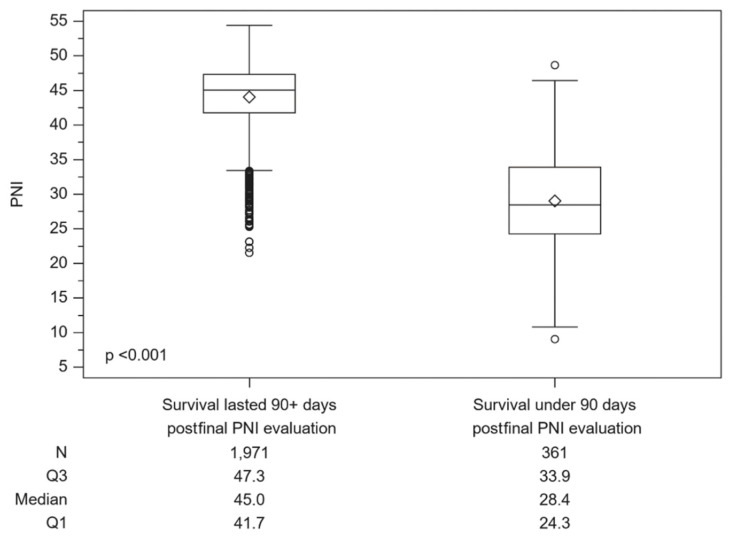
Box-plot of PNI distribution comparing groups with a survival time of more or less than 90 days post the final PNI evaluation. Mann–Whitney U test, *p* < 0.001.

**Table 1 cancers-16-01037-t001:** Comparison of patients’ characteristics and demographic features based on outcome status.

	Patients, No. (%) (N = 2332)		
Variable	Death (*n* = 638)	Alive (*n* = 1694)	*p*-Value ^a^
Age, mean ± SD (years)	54.4 ± 13.2	48.1 ± 11.3	<0.001
Sex			0.001
Male	510 (79.9%)	1245 (73.5%)	
Female	128 (20.1%)	449 (26.5%)	
T classification			<0.001
1	121 (19.0%)	631 (37.2%)	
2	108 (16.9%)	332 (19.6%)	
3	146 (22.9%)	377 (22.3%)	
4	263 (41.2%)	354 (20.9%)	
N classification			<0.001
0	65 (10.2%)	249 (14.7%)	
1	194 (30.4%)	714 (42.1%)	
2	210 (32.9%)	453 (26.7%)	
3	169 (26.5%)	278 (16.4%)	
*AJCC* Stage			<0.001
1	9 (1.4%)	114 (6.7%)	
2	72 (11.3%)	460 (27.2%)	
3	178 (27.9%)	552 (32.6%)	
4	379 (59.4%)	568 (33.5%)	
DM	44 (6.9%)	64 (3.8%)	0.001
HTN	75 (11.8%)	122 (7.2%)	<0.001
BMI (kg/m^2^)	24.5 ± 4.2	25.1 ± 4.0	0.001
Treatment protocol			0.519
IMRT	73 (11.4%)	169 (10.0%)	
CCRT	502 (78.7%)	1343 (79.3%)	
Induction C/T + CCRT	63 (9.9%)	182 (10.7%)	

SD = standard deviation; *AJCC* = American Joint Committee on Cancer Staging Manual, 7th and 8th Edition; DM = diabetes mellitus; HTN = hypertension; BMI = body mass index; IMRT = intensity-modulated radiotherapy; CCRT = concurrent chemoradiotherapy; and C/T = chemotherapy. ^a^ T-tests for continuous variables, and Chi-square tests /Fisher’s exact tests for categorical variables.

**Table 2 cancers-16-01037-t002:** Separate longitudinal analysis, with PNI as the response variable.

Variable	Estimate (95% CI)	SE	*p*-Value
Intercept	48.592 (46.934, 50.251)	0.846	<0.001
Follow-up (years)	−1.568 (−1.784, −1.352)	0.110	<0.001
Age (years)	−0.117 (−0.132, −0.101)	0.008	<0.001
Male	0.734 (0.311, 1.158)	0.216	<0.001
DM	−1.818 (−2.752, −0.885)	0.476	0.003
HTN	−0.274 (−0.994, 0.447)	0.367	0.170
BMI (kg/m^2^)	0.086 (0.040, 0.131)	0.023	<0.001
*AJCC* stage			
1	Ref.		
2	−2.137 (−3.248, −1.025)	0.567	0.001
3	−3.050 (−4.150, −1.950)	0.561	<0.001
4	−4.138 (−5.225, −3.050)	0.555	<0.001
Treatment protocol			
IMRT	Ref.		
CCRT	1.606 (0.843, 2.369)	0.389	<0.001
Induction C/T + CCRT	1.492 (0.597, 2.387)	0.457	<0.001

PNI = prognostic nutritional index; SE = standard errors; DM = diabetes mellitus; HTN = hypertension; BMI = body mass index; *AJCC* = American Joint Committee on Cancer Staging Manual, 7th and 8th Edition; IMRT = intensity-modulated radiotherapy; CCRT = concurrent chemoradiotherapy; and C/T = chemotherapy.

**Table 3 cancers-16-01037-t003:** Separate survival analyses using time-varying covariate Cox hazard model with overall survival (OS) as the event.

Variable	Hazard Ratio (95% CI)	SE	*p*-Value
PNI	0.813 (0.805, 0.821)	0.005	<0.001
Age (years)	1.011 (1.005, 1.018)	0.003	0.001
Male	1.445 (1.188, 1.759)	0.100	<0.001
DM	1.078 (0.764, 1.522)	0.176	0.669
HTN	1.269 (0.960, 1.678)	0.143	0.094
BMI (kg/m^2^)	0.985 (0.965, 1.006)	0.011	0.163
*AJCC* stage			
1	Ref.		
2	1.532 (0.741, 3.167)	0.371	0.250
3	2.549 (1.253, 5.185)	0.362	0.010
4	3.783 (1.873, 7.641)	0.359	<0.001
Treatment protocol			
IMRT	Ref.		
CCRT	0.813 (0.623, 1.060)	0.135	0.126
Induction C/T + CCRT	0.993 (0.694, 1.422)	0.183	0.971

PNI = prognostic nutritional index; SE = standard errors; DM = diabetes mellitus; HTN = hypertension; BMI = body mass index; *AJCC* = American Joint Committee on Cancer Staging Manual, 7th and 8th Edition; IMRT = intensity-modulated radiotherapy; CCRT = concurrent chemoradiotherapy; and C/T = chemotherapy.

**Table 4 cancers-16-01037-t004:** Joint modeling analysis of longitudinal PNI data and overall survival (OS). For distant metastasis-free survival (DMFS) and local–regional recurrence-free survival (LRRFS), please refer to the Appendix A.

Variable	Estimate (95% CI)	SE	*p*-Value
Longitudinal sub-model			
Intercept	48.592 (47.251, 49.934)	0.684	<0.001
Follow-up (years)	−1.568 (−1.613, −1.523)	0.023	<0.001
Age (years)	−0.117 (−0.129, −0.104)	0.006	<0.001
Male	0.734 (0.389, 1.080)	0.176	<0.001
DM	−1.818 (−2.516, −1.120)	0.356	<0.001
HTN	−0.274 (−0.832, 0.285)	0.285	0.337
BMI (kg/m^2^)	0.086 (0.053, 0.118)	0.017	<0.001
*AJCC* stage			
1	Ref.		
2	−2.137 (−3.126, −1.147)	0.505	<0.001
3	−3.050 (−4.036, −2.064)	0.503	<0.001
4	−4.138 (−5.107, −3.169)	0.495	<0.001
Treatment protocol			
IMRT	Ref.		
CCRT	1.606 (0.960, 2.252)	0.330	<0.001
Induction C/T + CCRT	1.492 (0.747, 2.236)	0.380	<0.001
**Variable**	**Hazard Ratio (95% CI)**	**SE**	***p*-Value**
**Survival sub-model**			
PNI	0.864 (0.850, 0.879)	0.009	<0.001
Age (years)	1.019 (1.012, 1.027)	0.004	<0.001
Male	1.390 (1.135, 1.702)	0.103	0.001
DM	1.398 (0.980, 1.993)	0.181	0.064
HTN	1.119 (0.834, 1.503)	0.150	0.453
BMI (kg/m^2^)	0.957 (0.936, 0.979)	0.012	<0.001
*AJCC* stage			
1	Ref.		
2	1.817 (0.909, 3.631)	0.353	0.091
3	3.525 (1.793, 6.929)	0.345	<0.001
4	6.286 (3.223, 12.262)	0.341	<0.001
Treatment protocol			
IMRT	Ref.		
CCRT	0.704 (0.531, 0.933)	0.144	0.015
Induction C/T + CCRT	0.660 (0.455, 0.959)	0.191	0.029

PNI = prognostic nutritional index; SE = standard errors; DM = diabetes mellitus; HTN = hypertension; BMI = body mass index; *AJCC* = American Joint Committee on Cancer Staging Manual, 7th and 8th Edition; IMRT = intensity-modulated radiotherapy; CCRT = concurrent chemoradiotherapy; and C/T = chemotherapy.

## Data Availability

Restrictions apply to the availability of these data. Data were obtained from the Chang Gung Research Database and are available with the permission of the Institutional Review Board (IRB) of the Kaohsiung Chang Gung Memorial Hospital.

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
