# Peer review of "A Joint Model Based on Post-Treatment Longitudinal Prognostic Nutritional Index to Predict Survival in Nasopharyngeal Carcinoma"

_cancers, 2024, doi:10.3390/cancers16051037_

Round 1

Reviewer 1 Report

Comments and Suggestions for Authors

This is a well-written paper by Hsiao and colleagues exploring the use of collaborative modeling for analysis longitudinal PNI data from each routine visit, examining their association with overall survival. 

            Noting the need to develop prognostic markers alternative to TNM that could provide a more individualized prediction of patient treatment outcomes, the authors proposed the Prognostic Nutritional Index. This indicator evaluates both nutritional status and immune status. 

               The Authors conducted a retrospective study of 2332 patients with NPC, analyzing the correlation between patient characteristics, including PNI, and overall survival. Joint modeling was conducted. The statistical analysis used ROC curve analysis and Youden's index.

The analysis showed that PNI may be a useful independent marker for overall prediction

survival of patients with NPC. Due to low costs, the authors recommend regular use

PNI measurements during each clinic visit. 

Tables and 3 figures are very informative and provide great data presentation. The Authors extensively discussed their results in relation to recent scientific literature and clinical data reports.  

Paper has 52 references which are relevant to article's subject.

            Taken together, this paper represents a worthwhile contribution to the NPC research. This is clinically valuable study. I recommend the manuscript for further publication process.

Author Response

We sincerely appreciate the time and effort you invested in reviewing our manuscript and are grateful for your positive and encouraging feedback. We are pleased to hear that our study is viewed as a valuable contribution to the field. Should there be any further requirements or recommendations to improve our manuscript, please inform us, as we are dedicated to making all necessary adjustments

Reviewer 2 Report

Comments and Suggestions for Authors

An interesting study to show emphasizes the importance of routine longitudinal Prognostic Nutritional Index (PNI) data in predicting survival outcomes for nasopharyngeal carcinoma (NPC) patients. There are some questions as below,

1) Firstly, the description of the background of NPC is too general in “1. Introduction”.We could add the new perspective now. A paper refreshingly proposes NPC ecology theory, and claim that NPC is pathological ecosystem with unity of ecology and evolution ( https://pubmed.ncbi.nlm.nih.gov/37056571/). This paper might provide a novel understanding of this disease.

2) “Previous research has predominantly focused on pre-treatment PNI” ( 1. Introduction), these previous mean Ref. [17-20]? The recent study about pre-treatment PNI (J Cancer Res Clin Oncol. 2023 Dec;149(20):17795-17805. doi: 10.1007/s00432-023-05485-5.) was omitted.

3) Why the PNI was defined as 10 × serum albumin (g/dL) + 0.005 × total lymphocyte count (count/μL)? Does this apply to NPC or all solid tumors?

4) I don't quite understand what this means “Each of PNI data was collected and calculated...starting from the point at which the NPC diagnosis was confirmed.”, it mean that pre-treatment PNI also included in this study at the beginning?  

5) In this study, the joint modeling approach was employed to estimate the relationship between overall survival (OS) and the longitudinal change of the PNI, how about progression-free survival (PFS), distant metastasis-free survival (DMFS) and localregional recurrence-free survival (LRRFS)? (Table 4, Figure 2)

6) “This study was approved by the institutional review board of Chang Gung Medical Foundation”, the approved number should be provided? 

Author Response

Q1:  Firstly, the description of the background of NPC is too general in “1. Introduction”. We could add the new perspective now. A paper refreshingly proposes NPC ecology theory, and claim that NPC is pathological ecosystem with “unity of ecology and evolution” ( https://pubmed.ncbi.nlm.nih.gov/37056571/). This paper might provide a novel understanding of this disease.

Response:
Thank you for providing this novel view on NPC. We will add it to the introduction and revise it as follows (Manuscript Lines 56 to 61):
“The pathogenesis of nasopharyngeal carcinoma (NPC) has been traditionally linked to Epstein-Barr virus (EBV) infection, genetic predispositions, and environmental factors [1]. Building on this, recent theories suggest framing NPC as a pathological ecosystem, where disease progression is influenced by a complex network of ecological and evolutionary interactions, involving the dynamic nature of the cancer and its capacity to adapt within the biological landscape it inhabits [3].”

Q2:  “Previous research has predominantly focused on pre-treatment PNI” ( 1. Introduction), these previous mean Ref. [17-20]? The recent study about pre-treatment PNI (J Cancer Res Clin Oncol. 2023 Dec;149(20):17795-17805. doi: 10.1007/s00432-023-05485-5.) was omitted.

Response:
Yes, we will include this recent study in our references. Thank you for bringing it to our attention.

Q3:  Why the PNI was defined as 10 × serum albumin (g/dL) + 0.005 × total lymphocyte count (count/μL)? Does this apply to NPC or all solid tumors?

Response:
This formula was originated by Onodera et al. in 1984, from a study assessing the nutritional status of 200 malnourished patients with cancer of the digestive system and its correlation with postoperative complications. The chosen coefficients—10 for serum albumin and 0.005 for lymphocyte count—were the result of a regression analysis that quantified the predictive association between preoperative nutritional status and postoperative outcomes. This index is applicable across a spectrum of solid tumors, not limited to NPC. As mentioned in Introduction of our original manuscript: “PNI is determined by assessing serum albumin levels and total lymphocyte count in the peripheral blood and has demonstrated prognostic value in various types of malignancies, including lung cancer, esophageal cancer, gastric cancer, colorectal cancer, pancreatic cancer, liver cancer, as well as head and neck cancer [12-19].”

We will incorporate the origin of the PNI, as established by Onodera et al., into the "Methods" section of our manuscript. The revised text (Manuscript lines 123 to 125) will read:

" The PNI was defined as 10 × serum albumin (g/dL) + 0.005 × total lymphocyte count (count/μL), with the formula originating from the work of Onodera et al. [40]. "

Ref:
Onodera, T.; Goseki, N.; Kosaki, G. [Prognostic nutritional index in gastrointestinal surgery of malnourished cancer patients]. Nihon Geka Gakkai Zasshi 1984, 85, 1001-1005.

Q4:  I don't quite understand what this means “Each of PNI data was collected and calculated...starting from the point at which the NPC diagnosis was confirmed.”, it mean that pre-treatment PNI also included in this study at the beginning?  

Response:
Yes, our study includes a set of PNI data collected as a pre-treatment measure at the time of diagnosis, allowing us to examine the impact of the longitudinal trajectory of PNI data on patient outcomes.

Q5:  In this study, the joint modeling approach was employed to estimate the relationship between overall survival (OS) and the longitudinal change of the PNI, how about progression-free survival (PFS), distant metastasis-free survival (DMFS) and local–regional recurrence-free survival (LRRFS)? (Table 4, Figure 2)

Response:
Apologies, we cannot provide results for progression-free survival (PFS) due to the lack of specific coding for disease progression in the Chang Gung Research Database. However, we are able to provide the results for distant metastasis-free survival (DMFS) and local–regional recurrence-free survival (LRRFS); please see the attachment.

It should be noted that longitudinal PNI was not identified as a statistically significant indicator for these two types of survival in the joint model analysis. These findings are presented in the 'Supplementary Material'

Q6:  “This study was approved by the institutional review board of Chang Gung Medical Foundation”, the approved number should be provided? 

Response:
The IRB approval number has been included in the manuscript at line 348. To further clarify and ensure transparency, we have revised the text in lines 101~102 to state: “This study was approved by the Institutional Review Board of Chang Gung Medical Foundation (reference number: 202001136B0C601).”

Reviewer 3 Report

Comments and Suggestions for Authors

Brief summary

The paper focus the attention on a very interesting topic, as lot of literature is present on correlation between nutritional status and patients prognosis in different types of cancer including head and neck malignancies

This is a clear paper, with structured and solid analysis of the parameters studied.

The results are reproducible, and the conclusions are consistent with the thesis and argument presented.

The conclusions are interesting and add advances in the current scientific knowledge.

No ethical problems are found in this study

I would like to make some suggestions and I have few questions

General concept comments

You can try to expand the discussion session.

The following papers includes cases of nasopharyngeal carcinoma, you can add it in the discussion

Fujiwara D, Tsubaki M, Takeda T, Miura M, Nishida S, Sakaguchi K. Objective evaluation of nutritional status using the prognostic nutritional index during and after chemoradiotherapy in Japanese patients with head and neck cancer: a retrospective study. Eur J Hosp Pharm. 2021 Sep;28(5):266-270. doi: 10.1136/ejhpharm-2019-001979. Epub 2019 Aug 17. PMID: 34426479; PMCID: PMC8403783.

Atasever Akkas E, Erdis E, Yucel B. Prognostic value of the systemic immune-inflammation index, systemic inflammation response index, and prognostic nutritional index in head and neck cancer. Eur Arch Otorhinolaryngol. 2023 Aug;280(8):3821-3830. doi: 10.1007/s00405-023-07954-6. Epub 2023 Apr 7. Erratum in: Eur Arch Otorhinolaryngol. 2023 Jun 8;: PMID: 37029321.

Moreover you can try to expand the discussion using other papers, as

Yang L, Xia L, Wang Y, et al. Low prognostic nutritional index (PNI) predicts Unfavorable distant metastasis-free survival in nasopharyngeal carcinoma: A propensity score-matched analysis. PLoS One 2016;11:e0158853. 8.

Du XJ, Tang LL, Mao YP, et al. Value of the prognostic nutritional index and weight loss in predicting metastasis and long-term mortality in nasopharyngeal carcinoma. J Transl Med 2015;13:364.

Luan CW, Tsai YT, Yang HY, Chen KY, Chen PH, Chou HH. Pretreatment prognostic nutritional index as a prognostic marker in head and neck cancer: a systematic review and meta-analysis. Sci Rep. 2021 Aug 24;11(1):17117. doi: 10.1038/s41598-021-96598-9. PMID: 34429476; PMCID: PMC8385102.

Comments on the Quality of English Language

english level is good 

Author Response

Thank you for your suggestion to broaden our discussion section with relevant literature. We recognize the significance of the study by Fujiwara et al. in enhancing our understanding of the prognostic value of the PNI during and after treatment for patients with head and neck cancer, including those with nasopharyngeal carcinoma (NPC). In response, we have updated our manuscript to include a discussion on the role of post-treatment PNI, citing the study by Fujiwara et al. along with other research.

Revised Section (Manuscript Lines 290 to 299):
“Currently, only four studies have explored the prognostic value of post-treatment PNI. The first study, involving 23 head and neck cancer patients undergoing concurrent chemoradiotherapy (CCRT), observed a correlation between PNI levels during treatment and the severity of mucositis. It further proposed utilizing post-treatment PNI as a criterion to guide discharge timing after completing CCRT [56]. The second study, which included 124 head and neck cancer patients—20 of whom had NPC—found that low post-treatment PNI levels significantly indicated poor prognosis [57]. The third study, specifically targeting NPC patients, linked lower PNI levels immediately post-treatment with poorer survival [27]. Similarly, the fourth study, assessing PNI one month post-treatment in NPC patients, confirmed this association.”

We also want to enroll other literature as references. For the literatures from Atasever et al. and Luan et al.:

Revised Section (Manuscript Lines 73 to 77):

“PNI is determined by assessing serum albumin levels and total lymphocyte count in the peripheral blood, and has demonstrated prognostic value in various types of malignancies, including lung cancer, esophageal cancer, gastric cancer, colorectal cancer, pancreatic cancer, liver cancer, as well as head and neck cancer [12-19].”

For the literature from Yang L et al. and Du XJ et al.:
Revised Section (Manuscript Lines 77 to 79):

“For patients with NPC, increasing evidence has demonstrated that a low PNI is associated with poorer survival outcomes in both the short-term and long-term [20-26].”

and line 245 to 247:

“The prognostic significance of PNI has been demonstrated by a series of studies, but the vast majority have relied solely on single time-point pre-treatment PNI measurements [20-26].”

Round 2

Reviewer 2 Report

Comments and Suggestions for Authors

The authors have fully answered my concerns. 

Reviewer 3 Report

Comments and Suggestions for Authors

The revised version is good, authors answered the questions and took the suggestions to improve the quality of the manuscript. 

Comments on the Quality of English Language

English level is good